# Multi-Model Running Latency Optimization in an Edge Computing Paradigm

**DOI:** 10.3390/s22166097

**Published:** 2022-08-15

**Authors:** Peisong Li, Xinheng Wang, Kaizhu Huang, Yi Huang, Shancang Li, Muddesar Iqbal

**Affiliations:** 1School of Advanced Technology, Xi’an Jiaotong-Liverpool University, Suzhou 215123, China; 2Data Science Research Center, Division of Natural and Applied Sciences, Duke Kunshan University, Suzhou 215316, China; 3Department of Electrical Engineering and Electronics, University of Liverpool, Liverpool L69 3BX, UK; 4School of Computer Science and Informatics, Cardiff University, Cardiff CF10 3AT, UK; 5Renewable Energy Laboratory, Communications and Networks Engineering Department, College of Engineering, Prince Sultan University, Riyadh 11586, Saudi Arabia

**Keywords:** edge computing, latency optimization, multi-model, task scheduling, autonomous driving, AI

## Abstract

Recent advances in both lightweight deep learning algorithms and edge computing increasingly enable multiple model inference tasks to be conducted concurrently on resource-constrained edge devices, allowing us to achieve one goal collaboratively rather than getting high quality in each standalone task. However, the high overall running latency for performing multi-model inferences always negatively affects the real-time applications. To combat latency, the algorithms should be optimized to minimize the latency for multi-model deployment without compromising the safety-critical situation. This work focuses on the real-time task scheduling strategy for multi-model deployment and investigating the model inference using an open neural network exchange (ONNX) runtime engine. Then, an application deployment strategy is proposed based on the container technology and inference tasks are scheduled to different containers based on the scheduling strategies. Experimental results show that the proposed solution is able to significantly reduce the overall running latency in real-time applications.

## 1. Introduction

The bandwidth congestion and heavy load on the core network always make traditional cloud computing unable to process data instantly and cause extra energy consumption. In order to combat latency in a real-time computing environment, a new dubbed edge computing paradigm is proposed, which is able to relocate the services originally hosted in the cloud server to the proximity of end devices [1].

Nowadays, Internet of Things (IoT) devices are usually equipped with abundant sensors and generate a large volume of data at the network edge [2]. However, it is often infeasible to transfer these massive data to the cloud because of the restricted network bandwidth and constraint reaction time [3]. As a result, there is a demand for artificial intelligence (AI) services to be deployed at the network edge, near where the data is generated. This demand has resulted in the convergence of edge computing and AI, culminating in a new paradigm—AI at the edge, also known as edge AI [4].

Edge AI is widely employed in the automotive industry [5]. For example, in Autonomous Driving (AD), AI in automotive can recognize dangerous situations by monitoring different sensors such as camera, light detection and ranging (LiDAR), and radio detection and ranging (RADAR) [6]. In this process, there are multiple tasks from different sensors that need different AI algorithms to run concurrently [7]. However, some problems hindering the development of autonomous driving must be addressed:

Firstly, much of the current autonomous driving technology development is focusing on improving target detection capabilities while ignoring considering how to reduce the overall system latency [8]. Autonomous driving systems must meet strict safety requirements and the vehicles should have the ability to control themselves autonomously as soon as possible [9]. So, one of the main challenges is that the task completion time must be low enough.

Secondly, the vehicles are typically equipped with heterogeneous computing hardware in order to achieve more processing capabilities and offer increasingly computation-intensive services at a lower cost [10]. For example, a system with traditional CPU elements can execute general computations; Graphics Processing Unit (GPU) and Field Programmable Gate Array (FPGA) can provide much more powerful computing capabilities for certain kinds of workloads in a cost-efficient way [11]. In this context, how to be compatible with heterogeneous platforms when deploying diverse AI models is challenging.

Thirdly, recent advances in deep learning techniques allowed significant improvements in the detection accuracy and running time of computer vision algorithms, accelerating their deployment in autonomous driving and industrial embedded systems [12]. Compared to cloud servers, embedded edge devices provide advantages such as low latency, low power consumption, low price, ease of deployment, etc. [13]. However, how to run multiple models concurrently on a single embedded device is still a new paradigm that is being explored.

In summary, the following challenges must be addressed:(1)*Running latency*. In many cases, the tasks are related and aim to collaboratively reach one goal instead of obtaining high quality in each standalone task. Therefore, one critical question in autonomous systems is how to reduce the overall running latency to meet the safety requirement.(2)*Hardware heterogeneity*. The computing devices are heterogeneous, consisting of CPUs, GPUs, FPGAs, and dedicated accelerators [14]. Therefore, the portability of AI models across different platforms is crucial.(3)*AI on embedded edge*. The DL model inference requires high memory and computational requirements. Fitting these algorithms onto embedded devices is a challenge in itself.

In order to address the aforementioned challenges, in this paper, we propose a scheduling strategy-based multi-model task execution system. Firstly, a runtime optimizer is designed to reduce the overall running latency by running multiple model inferences in a collaborative way. Secondly, the models trained on cloud servers are compressed and converted to ONNX, a cross-platform, high-performance inference engine for AI models trained from a variety of frameworks. Finally, a set of separately trained methods are held on embedded devices. Multiple apps are concurrently deployed on an embedded device using container technology and select an appropriate model to execute, respectively, based on the proposed runtime optimizer.

The main contributions are summarised as follows:(1)A real-time tasks scheduling strategy is proposed, in which multi-model tasks can be scheduled using a collaborative decision-making algorithm aiming to reduce the overall running latency without compromising the performance.(2)A DL model convert solution is proposed that can convert trained models from Tensorflow/Pytorch to ONNX to make an edge device able to concurrently run multiple DL workloads.(3)To address the heterogeneity of the edge computing system, a concurrent containerization scheme over the ONNX architecture is introduced for application.

In the next Section, we will introduce the related work.

## 2. Related Work

In this section, we reviewed the pioneer works on the development of running AI models on edge devices.

### 2.1. AI Inference from Cloud to Edge

Recently, in order to fully realize the potential of big data, there has been an urgent need to deploy AI services to the network edge [15]. To address this demand, edge AI has emerged as a promising solution, appearing in an increasing number of application scenarios [16,17,18].

Chen et al. [19] designed a Deep Learning-based Distributed Intelligent Video Surveillance (DIVS) system using an edge computing architecture. Gong et al. [20] proposed an intelligent cooperative edge (ICE) computing that combines AI and edge computing to build a smart IoT. In this paper, the generated data and AI model are distributed among IoT devices, edge and cloud.

However, in order to further improve the accuracy, Deep Neural Networks (DNNs) become deeper and with huge parameters to be processed [21]. The existing works focus mostly on higher accuracy, however, do not consider the hardware requirements. This causes a schism between software and hardware designs, resulting in algorithms that are highly accurate yet impossible to be implemented on embedded edge platforms with limited resources.

### 2.2. The Deployment of AI Model on Embedded System

Techniques for reducing the size and connectivity of AI model network architecture are attractive for deploying these models on embedded systems. Pruning [22] and quantization [23] are the two basic ways for reducing model memory footprint.

Minakova et al. [24] proposed using both task-level and data-level parallelism simultaneously for running Convolutional Neural Networks (CNN) inference on embedded devices. In this way, a high-throughput execution of CNN inference can be ensured and the restricted processing capabilities of embedded SoCs (Systems on Chip) can be fully used. Dey et al. [25] proposed using a partial execution strategy and partitioning algorithm for embedded systems to support scenarios that need to deal with heavy input data and huge model inferences in a short period of time.

Furthermore, TensorFlow Lite and TensorRT are considered cutting-edge inference compilers that incorporate most of the compiler optimization approaches offered for embedded devices. In [26], a detailed performance comparison of two recent compilers, TensorFlow Lite (TFLite) and TensorRT, is performed using commonly-used AI models on various edge devices.

However, the computing platform is heterogeneous and composed of various hardware components. In an autonomous vehicle, for example, the computing platform typically includes GPUs, CPUs, FPGAs and other dedicated deep learning accelerators [27]. Different frameworks for deploying AI models on hardware platforms can result in significantly different results.

### 2.3. Multi-Model Data Fusion

Nowadays, the fusion of multiple AI models has become a new paradigm. In [28], Mujica et al. introduced a novel messaging system-based edge computing platform to address the problem of how to efficiently fuse various data generated on heterogeneous hardware platforms. In [29], Fu et al. proposed fusing multiple roadside sensors, such as cameras and radar, on the edge of IoT networks to provide environment perception services for autonomous driving. In [30], an effective fusing approach for fusing the LiDAR and camera features is described, which is then processed for target detection and trajectory prediction. Mendez et al. [31] also designed a sensor fusion method integrating LiDAR and camera sensors for object detection that takes into account the vehicle’s limited computing capabilities while simultaneously reducing running latency by using edge computing architecture.

However, these methods only focus on precision while they did not consider how to reduce the processing latency.

### 2.4. Optimization of Latency

Edgent, proposed by Li et al. [4], employs edge computing for DNN model inference by collaborating device and edge. Model partitioning is utilized in this framework to separate computing tasks, and model right-sizing is used to further reduce latency by exiting inference at an appropriate intermediate DNN layer. In order to eliminate the time difference between data collection of sensors and approximately synchronize the data, Warakagoda et al. [32] only gather the data, including steering angle, LiDAR reading, and camera image, which are generated almost simultaneously. This approach solves the problem of the sensors operating at different frequencies. In [8], a multi-task environment detection framework is applied to autonomous driving with reduced time and power consumption. In this study, the vehicle detection model and lane detection model are combined to sense the vehicles’ surroundings and the weight pruning technique based on the Alternate Direction Method of Multipliers (ADMM) is utilized to decrease the running latency.

However, these studies focus on optimizing the inference time of a single AI model; time optimization for running multiple models on a single edge device is still a new paradigm that needs to be investigated.

This paper provides an effective solution to a difficult industrial challenge: Optimizing the edge computing architecture to reduce the latency when executing multiple model inferences on one edge device.

## 3. Problem Formulation

### 3.1. The Scenario

In the case of autonomous driving, for example, in order to better understand its surrounding environment, a vehicle must have multiple cameras mounted [33]. Depending on its function, each camera generates images at different frequencies, which can subsequently be processed and analyzed simultaneously by the vehicle’s object detection and recognition application. The architecture of the multi-model-based object detection system for autonomous driving is shown in Figure 1.

As indicated in Figure 1, appropriate deep learning models are selected for training in the Cloud server based on the vehicle’s object detection requirements as the cloud server has adequate computing capacity. The models trained in various frameworks are then converted to ONNX format. Each model is packaged into a container and deployed in the vehicle once it has been trained and converted into the cloud.

The procedure of object detection, as shown in Figure 2, can be divided into three stages: data collecting, data processing and information fusion. Firstly, three sensors (cameras) installed on the vehicle take pictures at a fixed frequency to acquire image data; then three different instances of the application (App)–one process for each camera–perform image-based object detection tasks independently with different AI models to detect vehicles, traffic lights and pedestrians in the images; finally, the system fuses the detection results to obtain the final execution instructions, which we call the collaborative decision making.

### 3.2. Problem Formulation

The executive duration of the task in each application (App) can be formulated as Equation (Equation 1).
(1)$$T_{task}=T_{in}+T_{pro}+T_{inf}+T_{out}$$
where $T_{in}$, $T_{pro}$, $T_{inf}$, and $T_{out}$ denote the time for inputting images, image processing, model inference and outputting result, respectively. Among them, the model inference accounts for the main part.

The running time of each App after the *n* rounds of tasks allocation can be formulated as Equation (Equation 2).
(2)$$T_{A}=\sum_{i=1}^{n}T_{task_{i}}$$

In this paper, we aim to reduce the accumulated overall multi-model running time. Because the collaborative decision-making occurs only after all the three tasks in three Apps in each batch have been completed, the overall running time is determined by the App that is the last to complete the *n* rounds of the task. In this case, the overall running time *T* can be calculated by Equation (Equation 3). The parameters are illustrated in Figure 3, the arrow with different colors represents different object detection tasks, and the length of the arrow denotes the task’s executive duration. In Figure 3, after the first three rounds of tasks allocation, the overall running latency *T* is $T_{A_{3}}$.
(3)$$T=\max\left\{T_{A_{1}},T_{A_{2}},T_{A_{3}}\right\}$$

However, this process does not easily allow for fast synchronization across multiple cameras. In order to make a final decision fast, we introduced the following scheduling strategies in Section 4.

## 4. System Design

This section introduces a typical scenario to apply the proposed method. Then, the designed two kinds of scheduling strategies are presented. The target is to minimize the average data fusion latency when multiple model inference workloads are running simultaneously on one edge device.

### 4.1. Optimal Selection Method

The first allocation strategy we have designed is the Optimal Solution Selection Method (OSSM). By the method, the optimal allocation solution is chosen each time, so that the overall latency can be guaranteed to be minimal. The method is described in detail as follows:

Firstly, there are *n* sensor inputs connected to one embedded edge device.
(4)$$S=\left\{s_{1},s_{2},\cdots ,s_{i},\cdots ,s_{n}\right\}$$
where $s_{i}$ represents the *i*th sensor input, *n* denotes the number of sensors.

Moreover, there are *n* Apps deployed at each edge device.
(5)$$A=\left\{a_{1},a_{2},\cdots ,a_{i},\cdots ,a_{n}\right\}$$
where $a_{i}$ represents the *i*th App.

For *n* inputs, each input task requires different processing and inference latency.
(6)$$L=\left\{l_{1},l_{2},\cdots ,l_{i},\cdots ,l_{n}\right\}$$
where $l_{i}$ represents the inference latency of $s_{i}$. The latency to complete a single inference task on a specific device is a basically fixed value.

Since the *n* tasks need to be assigned to *n* Apps, so there are n! assigning choices. the assigning matric is shown in Equation (Equation 8).
(7)$$C=\left\{c_{1},c_{2},\cdots ,c_{i},\cdots ,c_{n!}\right\}$$
where $c_{i}$ represents the *i*th allocation policy.
(8)$$\begin{array}{ccc} \; & \begin{array}{cccc} \;a_{1}\;\;\; & a_{2}\;\; & \cdots & a_{n} \end{array} & \\ \begin{array}{c} c_{1}\\ c_{2}\\ \vdots \\ c_{n!} \end{array} & \left[\begin{array}{cccc} s_{1} & s_{2} & \cdots & s_{n}\\ s_{2} & s_{1} & \cdots & s_{n}\\ \cdots & \cdots & \cdots & \cdots \\ s_{n!} & s_{n-1} & \cdots & s_{1} \end{array}\right] & \begin{array}{c} T_{c_{1}}\\ T_{c_{2}}\\ \vdots \\ T_{c_{n!}} \end{array} \end{array}$$

As shown in Equation (Equation 8), in each round of requests the *n* tasks will be assigned to different apps. Firstly, we can try each assigning choice and obtain the $T_{a_{i},c_{k}}$ by Equation (Equation 9).
(9)$$T_{a_{i},c_{k}}=T_{a_{i}}\left(r-1\right)+l_{j},\;i,j=1,2,\cdots ,n;\;k=1,2,\cdots ,n!$$
where *r* represents the *r*th round of tasks assigning, $T_{a_{i},c_{k}}$ is the accumulated running time of App $a_{i}$ under the allocation policy $c_{k}$.

Then, we can obtain the accumulated latency of each app under one specific choice by Equation (Equation 9). The data fusion latency depends on the app with maximum latency, so the data fusion latency of each assigning choice can be calculated using Equation (Equation 10).
(10)$$T_{c_{k}}=\max\left\{T_{a_{1}},T_{a_{2}},\cdots ,T_{a_{n}}\;|\;c_{k}\right\}$$
in which $T_{c_{k}}$ represents the accumulated running time under the *k*th allocation policy in the *r*th round of request.

In order to minimize the running latency, we choose the assigning $c_{x}$ with the minimum latency by Equation (Equation 11).
(11)$$c_{x}=\min\left\{T_{c_{1}},T_{c_{2}},\cdots ,T_{c_{n!}}\right\}$$

All *n* tasks can be allocated to *n* different apps based on the assigning choice $c_{x}$, and the accumulated latency of each app can be calculated by Equation (Equation 12).
(12)$$T_{a_{i}}\left(r\right)=T_{a_{i}}\left(r-1\right)+l_{k\;|\;c_{x},a_{i}}$$
(13)$$T_{a_{i}}\left(r\right)=\sum_{m=1}^{r}l_{k|m}$$
in which $T_{a_{i}}\left(r\right)$ represents the accumulated running time of *i*th application at the *r*th round of request.

The pseudocode of this process is shown in  Algorithm 1:
**Algorithm 1** Scheduling strategy 1**Input:** Initialization: *L*, *A*, $T_{a_{i}}$**Output:** Selection of scheduling combination $c_{x}$ r←1 **for** 
i←1ton
**do**  $T_{a_{i}}\left(r\right)\leftarrow l_{i}$ **end for** **repeat**  r←r+1  **for** k←1ton! **do**   **for** i←1ton **do**    Update $T_{a_{i},c_{k}}$ based on Equation (Equation 9)   **end for**   Update $T_{c_{k}}$ based on Equation (Equation 10)  **end for**  Update $c_{x}$ based on Equation (Equation 11)  **for** j←1ton **do**   Update $T_{a_{j}}\left(r\right)$ based on Equation (Equation 12)  **end for** **until** Scheduling finished

As shown in Algorithm 1, in this case, the time complexity is:(14)T(n)=O(n!)

The space complexity is:(15)S(n)=O(1)

At each allocation, this strategy compares all the allocation solutions and selects the one that results in the minimal running latency; therefore, this strategy can definitely obtain the optimal solution and minimize the running latency of multi-model inference tasks. However, the disadvantage of this strategy is that it takes a long execution time for allocation strategy-making when running a large number of model inferences simultaneously. In order to tradeoff between the execution time of the allocation strategy and the execution time of multi-model inference tasks, a second strategy is proposed in Section 4.2.

### 4.2. Simplest Allocation Method

The second allocation strategy we have devised is the Simplest Allocation Method (SAM). This method ensures that both the overall latency and the complexity of the algorithm can be reduced. The method is described in detail as follows:

Firstly, the *n* sensor inputs are ranked from the maximum to the minimum:(16)$$S_{rank}=Rank\left\{s_{1},s_{2},\cdots ,s_{i},\cdots ,s_{n}\;|\;descending\right\}$$

At each round, the accumulated running latency of *n* apps is ranked from the minimum to the maximum:(17)$$T_{rank}=Rank\left\{T_{a_{1}},T_{a_{2}},\cdots ,T_{a_{n}}\;|\;ascending\right\}$$

At last, the *i*th task in the $S_{rank}$ is assigned to the *i*th app in the $T_{rank}$:(18)$$T_{rank}\left(r\right)=T_{rank}\left(r-1\right)+S_{rank}$$

The pseudocode of this process is shown in  Algorithm 2:
**Algorithm 2** Scheduling strategy 2**Input:** Initialization: *L*, *A*, $T_{a_{i}}$**Output:** Selection of scheduling combination $c_{x}$ r←1 **for** 
i←1ton
**do**  $T_{a_{i}}\left(r\right)\leftarrow l_{i}$ **end for** Update $S_{rank}$ based on Equation (Equation 16) **repeat**  r←r+1  Update $T_{rank}$ based on Equation (Equation 17)  **for** i←1ton **do**   $T_{a_{i}}\left(r\right)\leftarrow T_{a_{i}}\left(r-1\right)+S_{rank}\left\{i\right\}$  **end for** **until** Scheduling finished

As shown in Algorithm 2, in this case, the time complexity is:(19)T(n)=O(nlogn)

The space complexity is:(20)S(n)=O(1)

The mathematical modeling of the strategy is shown in Figure 4. As shown in the figure, at each allocation, the task requiring the longest execution time is assigned to the currently idle application, until the task requiring the least execution time is assigned to the currently occupied application. This strategy can effectively reduce the overall running delay.

### 4.3. Overall System Description

The proposed optimized multi-model task scheduling and execution mechanism, as well as the schematic, are shown in Figure 5 and Figure 6, respectively.

As shown in Figure 5, images captured by various sensors are buffered on the edge device. Based on the proposed allocation strategy, each application (App) dynamically invokes a different AI model, which then conducts an object detection task on the input image and outputs the detection result. Finally, the collaborative decision is made based on the fusion of the detection results.

Figure 6 compares the running time before and after the tasks scheduling strategy is implemented. Before adopting the scheduling strategy, each application continuously invokes a fixed model to perform a fixed object detection task. The collaborative decision-making (gray area in the figure) is performed only when all the tasks in three applications in each batch have been completed. Following the implementation of the scheduling strategy, each application dynamically performs different tasks based on the allocation decision. As shown in the figure, distributing multiple tasks to different applications at the same time based on the current state of each application efficiently reduces the time interval between two decision-making, achieving the goal of lowering the overall running time.

Figure 7 shows the optimized system architecture, which can be divided into four stages: *sensing, scheduling, detection* and *final decision-making*. At first, the cameras mounted on the vehicle capture images of the surrounding environment; Secondly, the images are assigned to different applications based on the scheduling strategy and the status of each application (App), this process is deployed on the *Runtime Optimizer*; Then, each application invokes a different AI model to process the image, perform the AI-based target detection task, and output the detection results; Finally, the detection results from each application are forwarded to the final decision-making block, where they are fused and being post-processed for decision making.

The role of the *Runtime Optimizer* is to receive requests, perform a scheduling strategy, and then allocate the requests to each application for inference. Since the inference time for each model differs, once all three models have been executed, the detection results are then fused. So the overall running latency (execution time) for final decision-making is determined by the model that has the longest inference time.

## 5. Experiment

In this section, we describe the experimental setup in detail and demonstrate the latency advantages of our approaches.

### 5.1. Experimental Setup

In order to measure the performance of concurrent workloads execution at edge devices, in our test-bed, MobileNet, ShuffleNet and SqueezeNet are deployed on NVIDIA Jetson nano 2 GB.

#### 5.1.1. Hardware

The *Nvidia* Jetson nano is chosen in the experiments as an embedded edge device. Technical Specifications (SPECS) of Jetson nano 2 GB is shown in Table 1.

#### 5.1.2. Software

Three deep learning models, including MobileNet, ShuffleNet and SqueezeNet, are used in the experiment to evaluate the performance of the proposed scheduling strategies.

(1)MobileNet: As a lightweight deep neural network, MobileNet models are very efficient in terms of speed and size and hence are ideal for embedded and mobile applications.(2)ShuffleNet: An extremely computation-efficient CNN model designed specifically for mobile devices with very limited computing power.(3)SqueezeNet: SqueezeNet models are highly efficient in terms of size and speed while providing good accuracy. This makes them ideal for platforms with strict constraints on size.

#### 5.1.3. Model Deployment

(1)Model inference is executed based on *ONNX Runtime* Environment. ONNX Runtime is a cross-platform inference and training machine-learning accelerator, which supports models from various deep learning frameworks and is compatible with different hardware [34].(2)Containerized app deployment. Each application served for model inference on edge devices is packaged and run as containers. Containerization technology can naturally shield hardware heterogeneity and bring great convenience to deployment and management.(3)In this experiment, only the CPU is used to perform the model inference task. On the one hand, CPUs are ubiquitous and can be more cost-effective than GPUs for running AI-based tasks on resource-constrained embedded edge devices. On the other hand, we proposed using the ONNX runtime framework for model inference, which uses CPU and can speed up the model inference and result in lower costs, faster response times, and a more portable algorithm.

In addition, the image acquisition and analysis tasks were executed 1000 times. In order to validate the benefits of our proposed method, we implement and compare two proposed strategies against the system without scheduling.

### 5.2. Performance Evaluation

For this experiment, the main evaluation metric is overall running time, while the CPU usage and memory usage are also monitored during task execution.

#### 5.2.1. Overall Running Time

Before applying the strategy, the inference time of each model is shown in Figure 8.

The overall running latency of each application is shown in Figure 9.

The overall running latency of performing data fusion is determined by the App with the longest running time. As shown in Figure 8, MobileNet inference takes the longest running time, so the latency of data fusion is determined by MobileNet. According to Figure 9, MobileNet takes a total of 267,848 ms to conduct 1000-times inference tasks, hence the overall running latency after executing 1000 tasks is 267,848 ms.

The overall running time of the three Apps based on strategy 1 and strategy 2 are shown in Figure 10 and Figure 11, respectively.

As can be seen from Figure 9, before the implementation of the scheduling strategy, the time gap among the three applications becomes increasingly bigger as the tasks were executed, which resulted in time wasting. After the scheduling strategy is implemented, the time gap between the three Apps is basically not much different. In addition, as shown in Figure 10 and Figure 11, the overall runtime latency is significantly reduced after the scheduling strategy is applied.

The comparison of the overall running time under the three conditions, without scheduling and two strategies, is shown in Figure 12.

As can be seen in Figure 12, compared to without scheduling strategy, Strategy 1 and Strategy 2 reduce the runtime latency by 16.7% and 14.3%, respectively. It can thus be demonstrated that adopting the scheduling mechanism efficiently reduces the overall running time of the system, resulting in a faster reaction time for vehicles.

#### 5.2.2. Model Inference Time

The inference time is one of the most crucial metrics when deploying an AI model in a commercial setting. Most real-world applications necessitate lightning-fast inference time, ranging from a few milliseconds to one second. Inference time is measured as the time difference between the arrival time of an image and the completion time of object detection. The inference time of the three models in three situations is shown in Figure 13.

As can be seen from Figure 7, due to the implementation of *Runtime Optimizer*, tasks need to be allocated based on the state of the applications before being executed, and then the application will then invoke the corresponding AI model to perform model inference based on the task’s requirements. Therefore, as shown in Figure 13, the inference time of a single task is slightly increased as compared to without scheduling.

#### 5.2.3. Cpu Usage

We use *Docker stats* to display a live stream of container(s) resource usage statistics, including CPU and memory usage.

CPU usage in three situations is shown in Figure 14.

As shown from Figure 14a–c, each of the three applications takes up approximately one-third of the CPU resources, indicating that the proposed system can make full use of the resources of the embedded hardware system.

#### 5.2.4. Memory Usage

Memory usage in three situations is shown in Figure 15.

As illustrated in Figure 15a–c, the memory usage varies between the different applications, which is related to the size of the model. As can be seen from Figure 15b,c, the memory footprint of all three applications is roughly the same at around 7%. We attribute these results to the fact that each application has essentially the same functionality and each application can call one of the three models. On the other hand, the memory footprint of the models has increased compared to before the scheduling strategy was adopted because each application has more functionality than before, and each application has to have the ability to perform reasoning for all three models.

#### 5.2.5. Inference Accuracy

The size and accuracy of models are crucial for deploying multiple models on resource-limited edge devices in an edge computing architecture. The model size and inference accuracy of the three models are shown in Table 2.

#### 5.2.6. Overhead Analysis

In summary, by adopting the scheduling strategy, the memory footprint is increased slightly but the runtime latency is reduced significantly.

## 6. Conclusions

This paper presents an overall running latency optimization solution for multi-model fusion. By adopting a task scheduling strategy, time-consuming tasks are assigned to applications with light workloads, thereby minimizing the overall latency of the system when handling multi-model data fusion. At the same time, the constrained platform resources for service deployment and the heterogeneity of hardware platforms can be better addressed by using the ONNX architecture and containerization technology. Experimental results also show that the approach can significantly reduce reaction time and hence improve system security. For future trends, the Deep Learning methods can be used to optimize the multiple model inference tasks allocation in Edge computing architecture. For future work, we plan to use the Reinforcement Learning (RL) method to optimize the multi-model tasks allocation strategy, which could dynamically allocate model inference tasks to different applications based on the application state and task characteristics through autonomous learning.

## Figures and Tables

**Figure 1 sensors-22-06097-f001:**
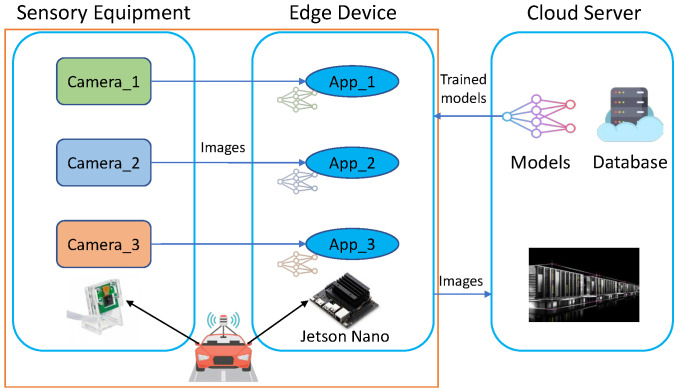
The architecture of the multi-model-based object detection system for autonomous driving.

**Figure 2 sensors-22-06097-f002:**
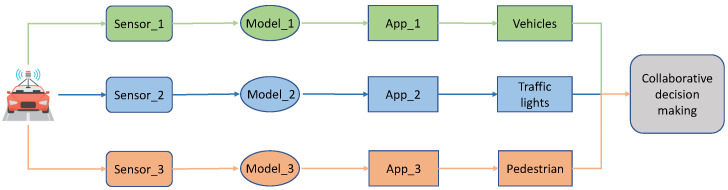
Data processing procedure of object detection and surrounding environment perception in autonomous driving.

**Figure 3 sensors-22-06097-f003:**
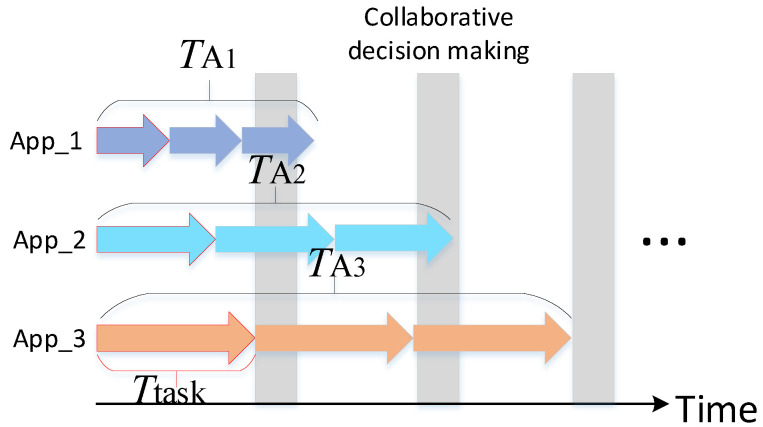
Schematic of tasks execution on the embedded edge device.

**Figure 4 sensors-22-06097-f004:**
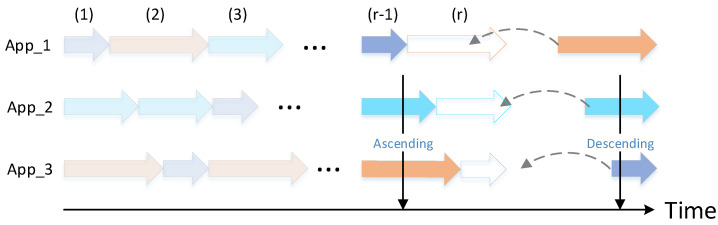
The mathematical modeling of the Simplest Allocation Method.

**Figure 5 sensors-22-06097-f005:**
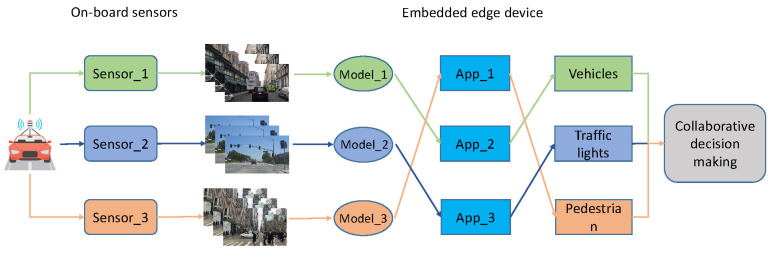
Optimized multi-model tasks scheduling-based object detection procedure.

**Figure 6 sensors-22-06097-f006:**
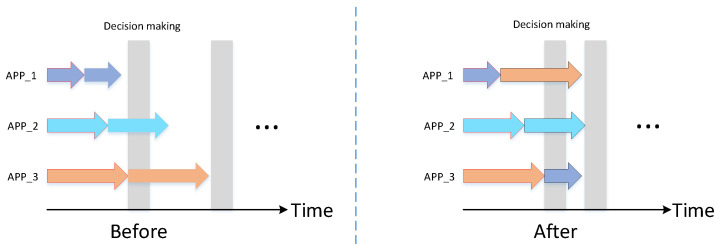
Comparison of the running time before and after the implementation of scheduling strategy.

**Figure 7 sensors-22-06097-f007:**
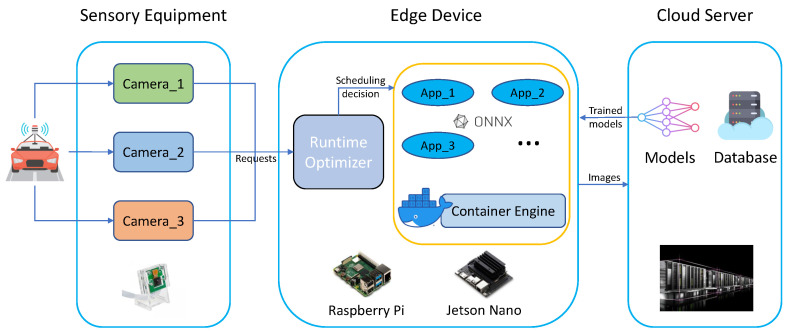
The edge computing architecture of the multi-model-based object detection system for autonomous driving.

**Figure 8 sensors-22-06097-f008:**
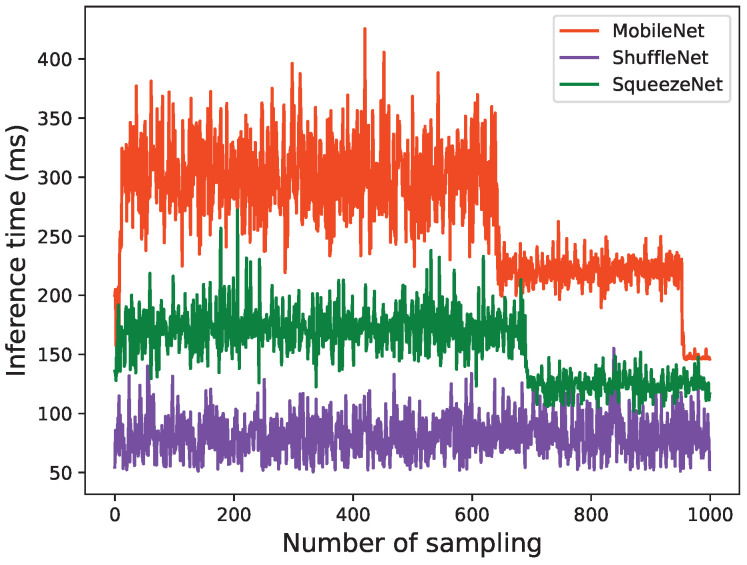
Inference time of each model without scheduling.

**Figure 9 sensors-22-06097-f009:**
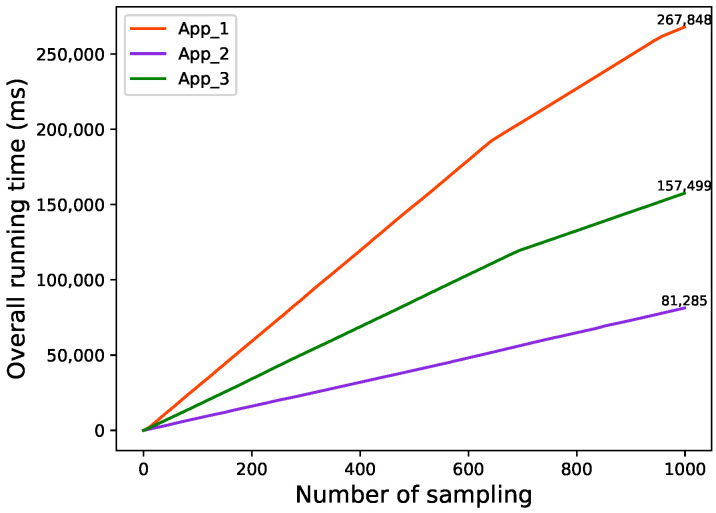
Overall running time of each App without scheduling.

**Figure 10 sensors-22-06097-f010:**
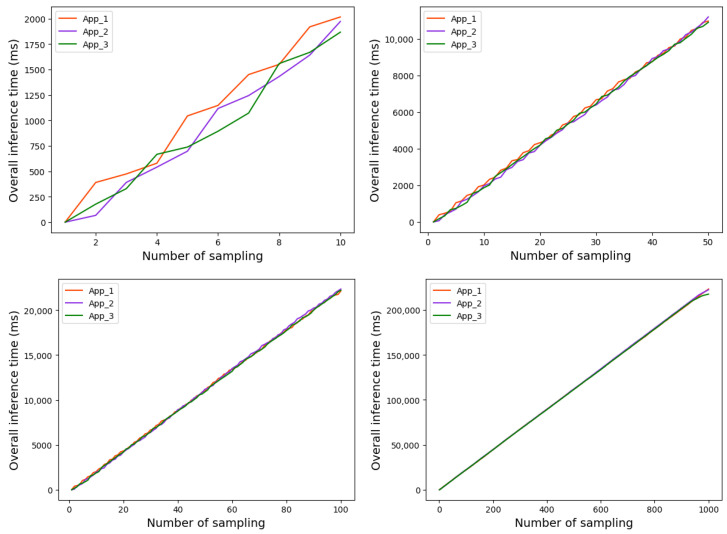
Overall running time of each App based on strategy 1.

**Figure 11 sensors-22-06097-f011:**
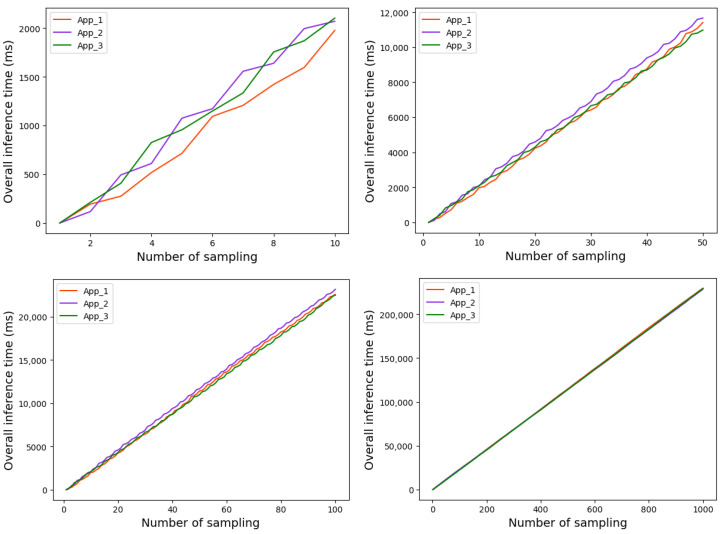
Overall running time of each App based on strategy 2.

**Figure 12 sensors-22-06097-f012:**
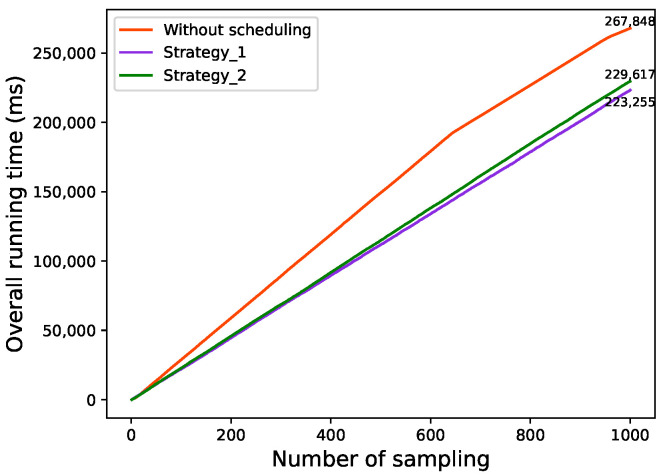
Comparison of overall running time under the three conditions.

**Figure 13 sensors-22-06097-f013:**
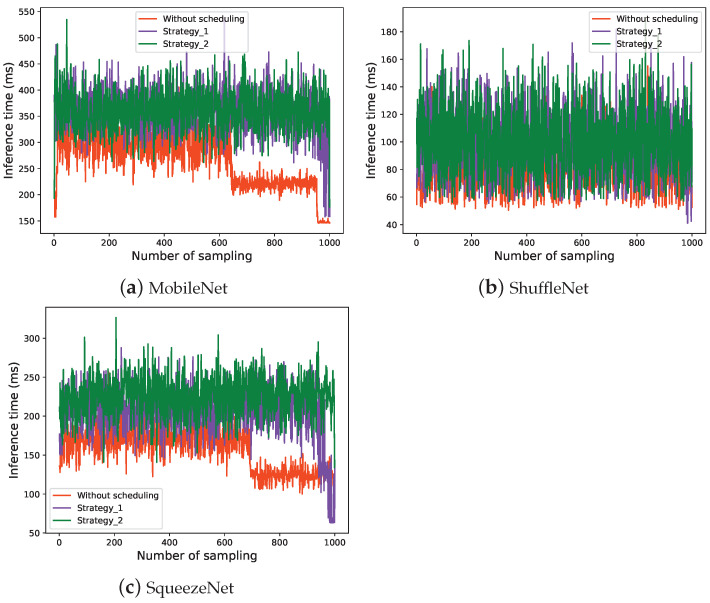
The inference time of the three models.

**Figure 14 sensors-22-06097-f014:**
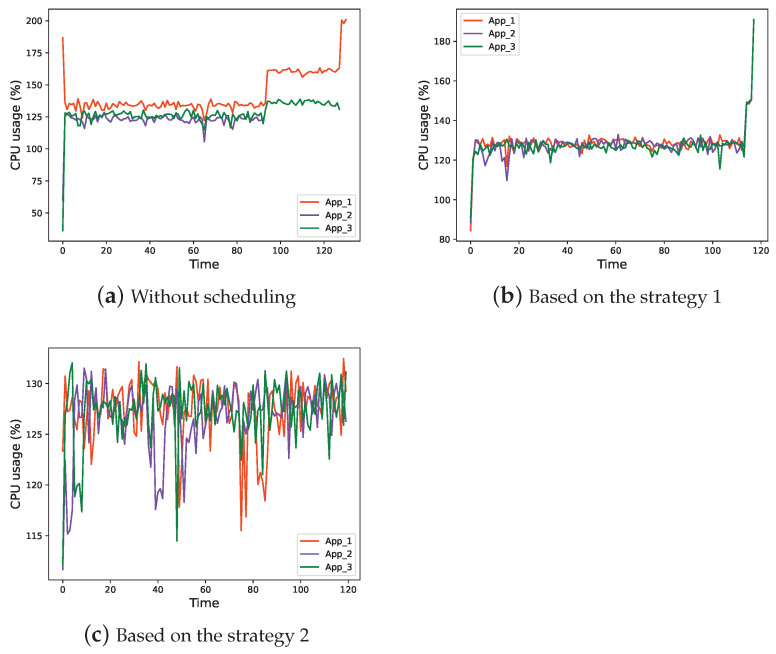
CPU usage of each App.

**Figure 15 sensors-22-06097-f015:**
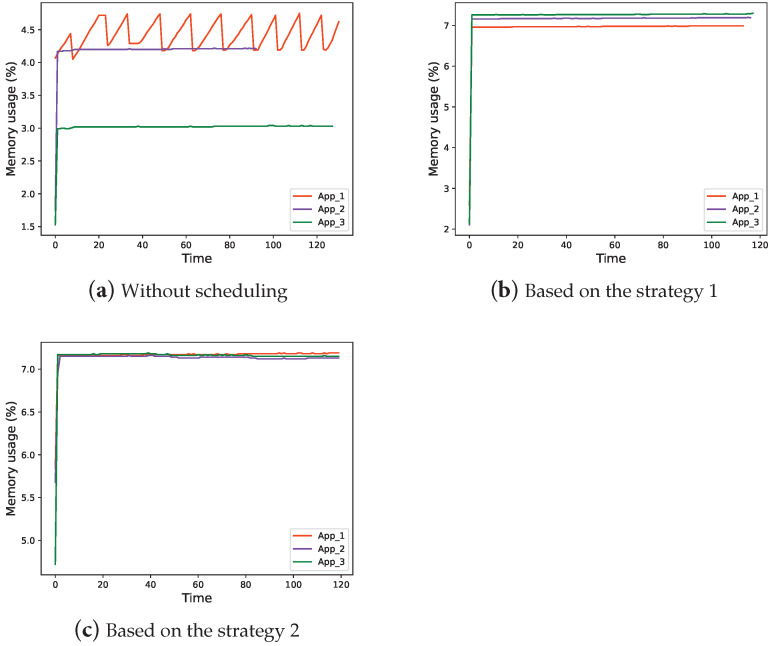
Memory usage of each App.

**Table 1 sensors-22-06097-t001:** Technical Specifications.

Technical Specifications	
GPU	128-core NVIDIA Maxwell
CPU	Quad-core ARM A57 @ 1.43 GHz
Memory	2 GB 64-bit LPDDR4 25.6 GB/s
Storage	64 GB

**Table 2 sensors-22-06097-t002:** Performance of the three models.

Model	Size	Top-1 Accuracy (%)	Top-5 Accuracy (%)
MobileNet	13.6 MB	70.94	89.99
ShuffleNet	9.2 MB	69.36	88.32
SqueezeNet	9 MB	56.34	79.12

## Data Availability

Not applicable.

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
