# Peer review of "Multi-Model Running Latency Optimization in an Edge Computing Paradigm"

_sensors, 2022, doi:10.3390/s22166097_

Round 1

Reviewer 1 Report

This research designs an optimized multi-model deployment scheduling algorithm to reduce the running delay from the perspective of practical application. The research has some certain application value. It is just that this research still has some unclear parts and needs to be improved. It is recommended that the authors refer to the following suggestions to revise the paper to meet the requirements of the paper being accepted for publication.

1. The multi-model running latency mentioned in the paper lacks a clear formal definition. It is unclear to the reader what components are included in the multi-model running latency? The authors should add clarifications.

2. Since the deep learning model trained in various model frameworks (TensorFlow, PyTorch, MXNET) is converted into a cross-platform model by the ONNX engine, it also requires model conversion processing time. It is unclear whether the two scheduling strategy algorithms proposed in this study consider the processing time factor of model conversion. The authors should add clarifications.

3. In addition, deploying the model converted by the ONNX engine into the container also requires model deployment time. Neither of the two proposed scheduling policy algorithms mentions whether this factor is taken into account. It’s unclear how the algorithm proposed in this study can really reduce the running latency of multiple models without considering the impact of model deployment containers. Please provide additional clarifications.

4. The two scheduling strategy algorithms proposed in this study also lack the analysis of time complexity and space complexity. In particular, the proposed scheduling algorithm will be applied to the edge computing network environment of autonomous vehicles, which has extremely high requirements on the computing time and storage space of the algorithm. Please provide additional clarifications.

5. The title of the paper is about the optimization of multi-model runtime latency in edge computing network environments, which is an interesting topic. However, the research to solve optimization problems is usually based on some optimization theoretical models. Although two scheduling algorithms are proposed, there is still a lack of mathematical model analysis to prove that the designed algorithm is optimal and can finally find the optimized solution, that is, the minimum running latency of multi-model deployment. Further clarification is suggested.

6. The description of Figures 2 and 3 is only a short sentence. Could the authors have more descriptions of these two figures to increase readers' understanding of the proposed method?

7. The vertical axis labels “overall inference time” in Figure 7 and Figure 8 lack units. Please revise them. 

8. The performance experimental analysis results of the paper are not yet comprehensive. Although the authors provide an analysis of the effect of sample size on the overall running time of multiple models. However, from a practical point of view, readers are more interested in the impact of the scheduling algorithm designed on the inference time and inference accuracy of multiple models. Especially in the edge computing network environment of the Internet of Vehicles for autonomous driving, the inference time requirements of various algorithms and models are at the microsecond level. The current performance analysis results of the paper are not enough to prove the effectiveness of the designed method. Additional clarification is recommended.

9. Academic papers should try not to use contractions (e.g. didn't). Contractions are used in informal spoken language; they are not used much in formal written English. On page 3, line 135, the sentence “However, these methods only focus on the precision while didn’t consider how to reduce the processing latency.” It is suggested that the contractions be changed to formal written English.

Author Response

Dear Reviewer,

Thank you for very much for your time and effort to very detailed and useful feedback on our manuscript. We have incorporated the suggested changes which are highlighted within the manuscript. We have attached a pdf that carries  point-by-point response to each of your comment in blue.

We look forward to your review and feedback on the revised submission. 

Kind regards

Authors

Reviewer 2 Report

I think that the work is interesting and opens several perspectives for study and experiment. I have the following observations:
1. The first sentence in the abstract "Recent advances in both lightweight deep learning algorithms and edge computing increasingly enable multiple model inference tasks to be conducted concurrently on recourse-constrained devices to collaboratively achieve a goal instead of standalone tasks" is very difficult to read and understand. Instead of “recourse-constrained” did you mean perhaps "resources - constrained"? Rephrase the sentence!

2. Add "ms" to the vertical labels in figures 7 and 8.

3. What is the meaning of CPU usage values greater than 100% in figures 10, 11 and 12? Explain.

4. Enter "future trends" in the conclusion section - future development directions starting from this article.

5. Why does the analysis in section 4 contain the CPU compute resource usage but not the GPU? The three applications should also use the GPU resource. Explain in section 4 or introduce analysis using GPU.

6. The Raspberry PI in Figure 4 illustrates that you also used this ePC for testing? Explain.

Author Response

(The authors gave the same response as above.)

Reviewer 3 Report

attached

Author Response

Dear Reviewer,

Thank you for very much for your time and effort to very detailed and useful feedback on our manuscript.  Your findings and comments have provided great insight by highlighting both strength and weakness of our manuscript. 

Thank you very much for encouraging response and recommendation. We agreed with pointed weakness and will use English Language expert services to proofread the camera-ready submission to ensure that paper language is understandable to general audience

Once again, thank you very much for your great insight and feedback. 

Kind regards

Authors

Round 2

Reviewer 1 Report

The author has responded positively to my proposed revisions one by one, and the revised manuscript has been significantly improved and has met the requirements of the paper being accepted for publication.